# The Role of Autoimmune Diseases in the Prognosis of Lymphoma

**DOI:** 10.3390/jcm9113403

**Published:** 2020-10-23

**Authors:** Pierluigi Masciopinto, Grazia Dell’Olio, Rosa De Robertis, Giorgina Specchia, Pellegrino Musto, Francesco Albano

**Affiliations:** 1Department of Emergency and Organ Transplantation (D.E.T.O.), Hematology and Stem Cell Transplantation Unit, University of Bari “Aldo Moro”, 70100 Bari, Italy; pierluigi.masciopinto@uniba.it (P.M.); g.dellolio05@gmail.com (G.D.); rosaderobertis88@gmail.com (R.D.R.); pellegrino.musto@uniba.it (P.M.); 2University of Bari “Aldo Moro”, 70100 Bari, Italy; specchiagiorgina@gmail.com

**Keywords:** autoimmune disease, lymphoma, prognosis

## Abstract

The connection between autoimmune disease (AID) and lymphoproliferative disorders is a complex bidirectional relationship that has long been a focus of attention by researchers and physicians. Although advances in pathobiology knowledge have ascertained an AID role in the development of lymphoproliferative diseases developing, results about AID influence on the prognosis of lymphoma are discordant. In this review, we collect the most relevant literature debating a direct or indirect link between immune-mediated diseases and lymphoma prognosis. We also consider the molecular, genetic, and microenvironmental factors involved in the pathobiology of these diseases in order to gain a deeper understanding of the nature of this link.

## 1. Introduction

Lymphomas are a heterogeneous group of lymphoproliferative diseases with differences in lymphocyte origin, biological development, clinical signs, and outcome. In fact, they arise from lymphocytes at various stages of development, and the characteristics of the specific lymphoma subtype reflect those of the cell from which they originated [1].

Autoimmune disease (AID) emerges from an aberrant immune system response to self-antigen, that damages tissues and organs. In many cases, the etiology of these pathologies is unclear, although it is assumed to be multifactorial with both genetic and environmental contributions. It is essential to distinguish AID arising from B and T cells. The first group includes myasthenia gravis, rheumatoid arthritis (RA), systemic lupus erythematosus (SLE), and Sjögren syndrome (SS); the second diseases such as dermatopolymyositis, inflammatory bowel diseases, multiple sclerosis, sarcoidosis, and type 1 diabetes mellitus [2].

The association between AID and lymphoproliferative disorders is now well known, even if the biological causes have not yet been entirely clarified. In this context, the most studied association is that with chronic lymphocytic leukemia (CLL) [3]. Several epidemiological studies have shown an increased risk of lymphoma associated with AID (e.g., SS, RA, SLE, celiac disease, etc.) [4], and in particular, an association between AID mediated B cells and non-Hodgkin lymphoma (NHL) (primarily, diffuse large B cell lymphoma) and between AID-mediated T cells and T-cells lymphomas [5]. In this review, we mainly focus on AID as a prognostic factor in patients affected by lymphoma without considering CLL, because, in this latter lymphoid disease, the prognostic role is clear.

The distribution of AID within NHL subtypes shows a high correlation between SS and B-cell NHL (in particular, the most frequent is diffuse large B-cell lymphoma (DLBCL)) and systemic vasculitis, celiac disease and T-cell lymphoma, SS and marginal zone lymphoma (MZL), SS, AHIA and angioimmunoblastic T-cell lymphoma (AITL) [6]. In Hodgkin lymphoma (HL), RA, and SLE, were the most common AID [7]. However, discordant results have been reported regarding the lymphoma subtypes most frequently associated with AID. In a case-control study among NHL adult patients, AID was most common in MZL (23.1%), followed by DLBCL (22.5%) and follicular lymphoma (FL) (20.7%) [8].

## 2. Pathophysiology of the Connection between AID and Lymphoma

A clear link between the inflammatory activity cumulative burden and the lymphoma risk has been demonstrated [9]. In this context, a crucial role is played in the inflammatory process and its self-maintenance by cytokines such as interleukin-2 (IL-2), IL-5, IL-6, IL-10, and tumor necrosis factor-α (TNF-α). Specifically, IL-10 is relevant because the presence of high levels of this cytokine can alter the delicate balance existing between T helper 1 (Th1) and 2 (Th2) lymphocytes. In this regard, Khanmohammad et al. [10] showed that the presence and contribution of cytokines are essential, especially for specific gene polymorphisms.

On the other hand, autoimmune conditions are more frequent during the lymphoma course (e.g., autoimmune cytopenias). Although this has been largely confirmed, the pathogenic mechanisms of this association are unclear. It has been suggested that antigen stimulation and chronic inflammation during autoimmune disease could expose the lymphoid cell to a higher risk of genetic events, leading to clonal expansion and, consequently, lymphoma development [11]. Other theories focus on the role of immunosuppressive therapies and environmental factors [12]. In detail, the use of steroids could not only have a role in lymphoma onset, but also a negative impact on disease prognosis (e.g., the use of steroids could retard lymphoma diagnosis, hiding the symptoms). In the case of AID onset during the lymphoma course, literature data suggest that impaired cellular and humoral-mediated immunity, which are often present in patients with lymphoma, and anti-red cells and platelets antibodies production by lymphoma cells, might promote the development of autoimmune alterations in this population [5,13].

Moreover, in an interesting study by Kuksin’s group [14], a primary role of the NOTCH1 and NOTCH2 pathways, that are usually deregulated in both diseases, was hypothesized. NOTCH receptor is involved in numerous cellular processes, including T cells maturation, so its dysfunction could affect both processes. In cases where the NOTCH pathway is chronically active, a permanent inflammatory state is created that can damage the immune system and determine oncogenic events, triggering the lymphomagenesis process (Figure 1).

It has also been proposed that the Fas receptor could be involved in both the development of AID and of lymphoproliferative diseases since, under physiological conditions, it is responsible for triggering the apoptosis process of self-reactive B and T lymphocytes. Therefore, a down-regulation of the Fas gene would make the self-reactive cells resistant to this process [10] (Figure 1).

A profound link between AID and lymphoma could also be via clonal hematopoiesis (CH). CH is a phenomenon caused by the acquisition of one or more mutations in hematopoietic stem cells (HSCs) that confers a replicative advantage; thus, in these circumstances, a population of mutated mature blood cells is produced, defined as a clone. When this clone reaches 2% of variant allele frequency, detected by next-generation sequencing technologies, the phenomenon is defined as CH of indeterminate potential (CHIP). There is growing evidence of a correlation between CHIP and the development of hematological malignancies (mainly myeloid) and other non-neoplastic diseases [15].

Apart from gene mutations, the primary mechanism involved in the development of CHIP seems to be inflammation; in fact, there is evidence of an association with high levels of cytokines, such as IL-6, IL-8, and TNF-α. Another significant association is with age, which could be explained both by an increased risk of gene mutations as HSCs proliferate, and by a phenomenon called “inflammaging,” namely a state of low-grade inflammation caused by modifications of cytokines production (mainly pro-inflammatory) occurring together with the aging process [15]. The relationship between CHIP and inflammation seems to be bidirectional. There is evidence that, in the context of CHIP, mutated-HSCs have a superior replicative capacity and resistance to apoptosis compare to wild-type HSCs when exposed to inflammatory stimuli. This could cause an increased pool of mutated circulating leukocytes and an abnormal production of pro-inflammatory cytokines that could act on mutated-HSCs and further promote clone expansion. These events suggest that AID, through inflammation, could boost CHIP development resulting in the production of mutated lymphocytes that could eventually cause lymphoproliferative disorders but also that CHIP could be the “primum movens” of both the impaired immune function that leads to AID and lymphomagenesis (Figure 1). Husby et al. [16] demonstrated an association in lymphoma patients undergoing autologous stem cell transplantation who presented CHIP, with a mutation in the DNA repair gene pathway of (such as PPM1D, TP53, RAD21, and BCRCC3) and a worse late overall survival (OS) and event-free survival. In detail, 80% of patients who presented this condition died during the follow-up. In more than half of them, death was caused by adverse events, such as infection and the development of a therapy-related neoplasm. It is interesting to note that the association between CHIP and inflammation is not confirmed for every inflammatory condition, e.g., RA, which could indicate a potential role of anti-inflammatory drugs in suppressing CH [17]. Therefore, further examination of the complex biological processes (responsible for both conditions) could lead to a better understanding of the clinical pathologies, thus increasing the possibility of adopting a specialized medicine approach.

## 3. Timing of AID in the Lymphoma Course

It is known that AID can arise before, during, or after the diagnosis of NHL. In a large retrospective study, it was demonstrated that the diagnosis of AID was antecedent to lymphoma onset in 65% of patients, concomitant in 27% of cases, and subsequent in 7% of patients. Furthermore, it was found that psoriasis and RA were mostly pre-existing conditions. In contrast, the AID diagnosed more frequently during or after lymphoma were immune cytopenias, such as autoimmune hemolytic anemia (AIHA), immune thrombocytopenia (ITP), and SS [18]. It is noteworthy that NHL patients with pre-existing AID diagnosis and a previous steroid therapy had a worse survival [6]. Finally, it has been reported that in NHL autoimmunity preceded the lymphoma diagnosis, but in HL the autoimmunity developed mainly after the treatment of malignancy [19].

## 4. Lymphoma, AID, and Outcome

It has been reported that specific autoantibodies can have a role in the pathogenesis and prognosis of lymphoma. In fact, 26% of patients with aggressive NHL, and 38% with HL were reported to have serum positivity to anti-phospholipid antibodies, and a worse prognosis [18]. 

Ludvingson et al. [20] analyzed the prognosis of a vast number of patients with lymphoma and celiac disease vs. patients affected by lymphoma only. It was demonstrated that the coexistence of the two affections defined a worse outcome, but only in the first year after lymphoma diagnosis. It emerged that the cause of this result was the lymphoma subtype more frequently associated with celiac disease (T-cell lymphoma). 

In a Swedish retrospective study [21], conducted in the pre-rituximab era, a set of 1523 patients affected by NHL was analyzed. Among them, 6% had a previous history of AID, principally AR, SS, SLE, and Hashimoto’s thyroiditis (HT); in this subset, at diagnosis of the onset of lymphoma, male patients with more advanced disease were those most commonly affected. It was noteworthy that the worse outcome was due to an increased non-lymphoma-correlated risk of death, suggesting that patients with AID were more susceptible to serious, potentially life-threatening therapy-related side effects. 

In another study conducted during the rituximab era, a decrease in relapse-free survival (RFS) and OS was observed in patients affected by B-cell NHL and B-lineage related AID; moreover, AID was associated with a poorer OS and RFS in DLBCL [8]. 

In a retrospective study, 2884 patients were analyzed; among them, 240 (8.3%) had a concomitant AID. The patients whose well-characterized AID diagnoses, such as SLE, SS, and RA, had been made 10 years or more before the lymphoma onset were excluded. Among the 108 patients included, 71.3% had autoimmune cytopenias, 10.2% Guillain Barrè syndrome (GBS), or other neurological diseases, 6.5% kidney diseases, 5.6% systemic vasculitis. As compared with other lymphoma patients, the patients with AID were older, more frequently had CLL, and their 5-year OS was 65% (vs. 79%) [22].

The above results are in conflict with other reports in the literature; for example, Hu et al. [6] found no OS differences between NHL patients with and without AID, and that the only factor linked with a worse outcome of these patients was previous corticosteroid therapy. Another interesting study conducted by Mikuls et al. [23] analyzed the impact on survival of lymphoma that arose in the context of RA. In this report, all patients whose diagnosis of RA was subsequent to the onset of lymphoma were excluded from the analysis; finally, 65 cases of lymphoma with RA were identified and compared with 1530 non-RA controls matched for age, sex, lymphoma type, diagnosis-to-treatment lag time and calendar year of treatment (between 1984 and 2002). The OS was not statistically different between RA patients and controls, but RA patients presented a reduced risk of progression, relapse, and lymphoma-related death. This favorable impact on prognosis was offset by a two-fold increased risk of death from causes unrelated to lymphoma.

## 5. Outcome in Patients with HL and Pre-Existing AID

A very high incidence of AID associated with HL is described in the literature. A retrospective Swedish study [7] that assessed survival pattern in patients with HL with and without a history of pre-existing AID found a worse outcome in the former; in fact, the 5-year OS was 46% versus 63.3%, and the 10-year OS was 41% versus 51.9%. The study also found that the presence of AID in HL patients had more influence on females compared to males (excess relative risk of death was 1.8 versus 1.7). In this subset of patients, the most common causes of death were lymphoma for female patients (72%) and therapy-related complications for male patients (68%). Moreover, cardiovascular deaths were more common in patients with a history of RA. These results are in contrast with another study in which only patients with Bechet’s disease and pernicious anemia had an increased mortality due to subsequent HL [24]. Although autoimmune cytopenias are rare paraneoplastic events in HL patients, they are well described in the literature. In fact, it is known that AIHA and ITP can occur in all HL subtypes (the most frequent is mixed cellularity subtype), that they happen more frequently in advanced disease (stages III/IV and bone marrow involvement) and older HL patients [25]. Nonetheless, no differences in terms of survival came out from comparisons between HL patients with and without autoimmune cytopenias [26].

## 6. DLBCL and Autoimmunity

DLBCL is the most widespread NHL subtype in the world. It is known that, like other NHL subtypes, B-cell mediated AID is an essential risk factor for DLBCL onset, but the data concerning the role of their association in the DLBCL outcome are contradictory. Analyzing the DLBCL outcome influenced by concurrent AID, Koff et al. [27] demonstrated a decreased lymphoma-related mortality in patients with SLE and DLBCL compared with DLBCL patients without AID, but this difference was not statistically significant. In another retrospective study conducted by a Swedish study group [28], it emerged that DLBCL patients with AID mediated by B-cell activity (with the exclusion of thyroid disorders) had a worse OS, but this seemed to affect only women. Moreover, the AID group more often had neutropenic fever (NF) after the first treatment, and those with NF presented a worse OS. Other studies, conducted in the pre-rituximab era, demonstrated an association between the presence of RA and the specific DLBCL cell of origin (COO) type. In detail, it was shown that the majority of RA-related DLBCLs belonged to the non-germinal center (GC) subtype and that these DLBCLs were characterized by an advanced disease stage at presentation and a worse prognosis than DLBCLs of the GC subtype (5-year OS 16% versus 33%) [29].

## 7. Mantle Cell Lymphoma and Autoimmunity

Mantle cell lymphoma (MCL) is an aggressive lymphoid neoplasm that arises from mature B cells in GC of lymph nodes or bone marrow. Even though many NHL subtypes are frequently associated with autoimmunity phenomena, MCL rarely coexists with this latter; it is thought that the receptor editing mechanism could explain this discordance [30]. In literature, only the association between MCL, AIHA, and ITP are counted: it was observed that in the context of MCL, AIHA could develop more frequently in patients with indolent leukemic non-nodal disease, and suggested that this disease presentation could share some biological and pathogenic features with classical CLL, favoring the occurrence of hemolysis. Surprisingly, it was noted that concomitant AIHA in MCL is associated with early lymphoma progression, which is correlated with poor outcome [31]. However, the small number of cases reported does not allow us to reach any definite conclusion.

## 8. Marginal Zone Lymphoma and Autoimmunity

MZL is an indolent subtype of lymphoma, which includes three distinct entities: extranodal marginal zone of mucosa-associated lymphoid tissue lymphoma (ENMZL), splenic marginal zone lymphoma (SMZL), nodal marginal zone lymphoma (NMZL). It is strongly believed that these entities arise from chronic activation of the immune system, which is typical in some AID. In particular, the studies note the link between pre-existing SS, HT, SLE, and ENMZL of salivary glands, primary thyroid lymphoma, and ENMZL of MALT-type, respectively [3,32]. Regarding the association between MALT lymphoma and AID (such as SS, RA, and SLE) it was reported that the presence of this latter had no influences in the outcome of these patients in terms of overall relapse rate [33]. Conversely, hematological autoimmune manifestations, such as ITP and AIHA, are frequent complications that become ongoing during the lymphoma course. These manifestations are documented in about 10–15% of SMZL and ENMZL patients [34], in particular, it emerged that the connection between AHIA and splenic lymphoma with villous lymphocytes was strong and that the most of these patients had warm antibody-AHIA [35]. Very little is known about the prognosis of this association. Disease prognosis worsens with transformation to a high-grade lymphoma (OS at 5 years 44%); this occurrence is associated with high lactate dehydrogenase (LDH) levels, which could be secondary to hemolysis, low serum albumin, and the ongoing autoimmune cytopenias [36]. In addition, authentic autoimmune conditions have been linked to the use of rituximab as a single agent and also of nucleoside analogs. In particular, rituximab and fludarabine treatment are associated with late-onset neutropenia (LON) and AIHA, respectively. Therefore, it is possible that the use of these agents may further exacerbate existing immune alterations in MZL.

## 9. T- Cell Lymphoma and Autoimmunity

There are very little data regarding the role of AID in T-NHL prognosis. Nevertheless, many scientific works about their association have been published in literature. It is known that T-NHL lymphomas are usually associated with AID that ranges from autoimmune cytopenias to rheumatologic and cutaneous autoimmune disorders. This circumstance could be imputed to the increased B-cell activation resulting from an aberrant proliferation of follicular helper T cells [37]. In a multicentric retrospective study, it was demonstrated that patients with AITL and AID more frequently had spleen (54% vs. 19%) and bone marrow involvement (71% vs. 34%), together with higher gamma globulin titers (23 g/dL vs. 15 g/dL) than patients with AITL only [38]. However, despite these being negative poor prognostic factors, median OS was 77 months and median PFS was 12 months in the AITL with AID group versus 33 months and 11 months in the control group.

## 10. Discussion

Autoimmune conditions such as RA, LES, SS, and celiac disease are defined as a risk factor for the development of lymphoma. The lymphomagenesis mechanism is still unclear, but chronic activation of the immune system and immunosuppressive drug action is hypothesized to have a significant role. Nevertheless, further studies are needed to determine whether lymphoma arising in the context of an autoimmune condition could constitute a distinct biological subtype with a different outcome. As regards autoimmune manifestations diagnosed after lymphoma onset, they could be the expression of an impaired immune-response caused by the lymphoma, and their impact on prognosis could be related to treatment-tolerability, especially in the case of autoimmune cytopenias. In this regard, recently, a study demonstrated in NHLs a relationship between imbalances in helper/suppressor T-cell populations and the development of auto-antibody production after chemotherapies [39].

A new field of research is focused on a possible role of CHIP in lymphomagenesis, as well as in the development of autoimmune conditions. In view of the fact that CHIP constitutes a risk factor for the development of inflammation, in the future, it could be regarded as a surrogate biomarker in patients affected by AID diagnosed with lymphoma, and also as a possible therapeutic target. In conclusion, it is not yet possible to define autoimmunity as a prognostic factor of lymphoma, and the correlations mentioned above warrant further studies. Moreover, the results of the various studies presented are sometimes contradictory (Table 1).

However, it is necessary to take into account the impossibility to compare these data because of the different inclusion criteria of patients with AID, the different patients selection (e.g., by telephone interview or hospital discharge diagnosis), which could overestimate or underestimate the real presence of an autoimmune condition, and also the relative frequency of patients affected by both diseases. Another critical difference among the studies is the change that occurred in lymphoma treatment in the last decades, since the use of rituximab. This latter is also a treatment for some of the autoimmune conditions mentioned above and could partially explain the difference in OS noted between the more dated studies as compared with the recent ones.

## Figures and Tables

**Figure 1 jcm-09-03403-f001:**
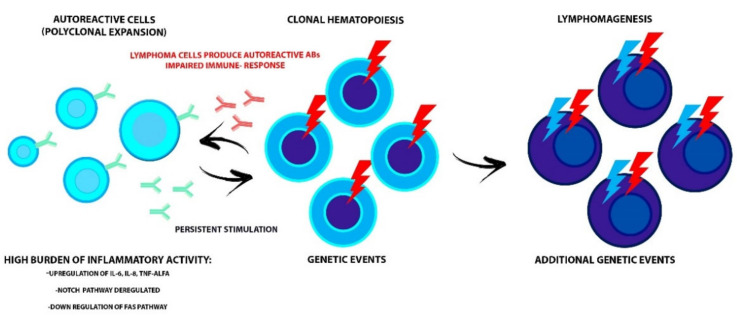
The interdependent relationship between AID and malignant clone evolution. The high burden of inflammatory activity could prepare for the acquisition of genetic events in HSC and progression to lymphoid malignancy. Conversely, the impaired immune-response that affects the lymphoproliferative disorders and the production of autoreactive antibodies by lymphoma cells leads to AID onset. Alterations of the NOTCH and FAS pathways, involved in the maturation and survival process of lymphocytes, could trigger both diseases. AID, autoimmune disease; HSC, hematopoietic stem cells; TNF-ALFA, tumor necrosis factor-alfa.

**Table 1 jcm-09-03403-t001:** AID role in the prognosis of lymphoma by subtypes.

Lymphoma Subtype	AID	Documented Role of Aid in Lymphoma Prognosis	Comments
HL	RA, SLE	Yes	HL patients with AID 5-y OS 46% (vs. 63.3%); 10-y OS 41% (vs. 51.9%). In female patients the most common cause of death was lymphoma (72% of cases) [13].
	Bechet’s disease, PA AHIA, ITP	No No	The mortality is increased due to lymphoma only in Bechet’s disease and pernicious anemia [24]. No difference in terms of OS came out from comparisons between HL patients with and without autoimmune cytopenia [26]
DLBCL	Thyroid disease RA, B-cell mediated AIDs	Yes	No difference in EFS or OS between the two groups, but female patients with primary AID (thyroid disorders excluded) had a worse OS [20]. Post-rituximab era.
	RA	Yes	The majority of RA-related DLBCLs belong to the non-GC subtype and had a worse prognosis than the GC subtype [29]. Pre-rituximab era.
	RA, SLE,SS	No	Low LRS in patients with SLE and DLBCL compared with other groups (but not statistically relevant) [5]. Pre-rituximab era.
MCL	Autoimmune cytopenia	No	The contingency of AIHA in MCL is associated with early lymphoma progression, which is correlated with poor outcomes. (too small sample size) [28].
MZL	RA, SLE, SS Autoimmune cytopenia	No No	The presence of AID has no significant influence on the long term clinical course of MALT lymphoma [33] The occurrence of autoimmune cytopenias during MZL is a sign of transformation to a high-grade lymphoma [30].

HL, Hodgkin lymphoma; RA, rheumatoid arthritis; SLE, systemic lupus erythematosus; AID, autoimmune disease; OS, overall survival; PA, pernicious anemia; DLBCL, diffuse large B-cell lymphoma; EFS, event-free survival; GC, germinal center; SS, Sjögren syndrome; MCL, mantle cell lymphoma; AIHA, autoimmune hemolytic anemia; MZL, marginal zone lymphoma; MALT, mucose-associated lymphoid tissue.

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
