# Peer review of "The Role of Autoimmune Diseases in the Prognosis of Lymphoma"

_jcm, 2020, doi:10.3390/jcm9113403_

Round 1

Reviewer 1 Report

GENERAL COMMENTS

The authors present a review on “The role of autoimmune diseases in the prognosis of lymphoma” but the content of the paper is not focused on the prognostic role of autoimmune disorders in lymphomas. The authors could improve on their writing and structure forming. Documentation of the presented information in terms of references is also poor with many important studies having been omitted. Some specific comments are given below but the concerns regarding this paper is not limited to them.

SPECIFIC COMMENTS

  1. The manuscript is composed by separate unrelated paragraphs but does not follow a clear structure and a rational order of reporting.
  2. The introduction is too long. Its content should be split in separate paragraphs under separate headings or subheadings. Pathophysiology should be discussed separately.
  3. Paragraphs 2 and 3 have extremely limited documentation.
  4. Many associations of significant lymphoma subtypes are not adequately described (autoimmune cytopenias and CLL, autoimmune cytopenias and Hodgkin lymphoma, pure red cell aplasia and CLL, marginal zone and mantle cell lymphoma etc).
  5. Some data appear rather peculiar. For example the 50% frequency of anti-phospholipid antibodies in the serum of aggressive non-Hodgkin lymphomas sounds rather contradictory to common clinical experience and reference #18, which is cited to support this, appears unrelated.

Reviewer 2 Report

This is the review of autoimmune diseases (AID) in the lymphoma. AID can occur before, during, or after the treatment of NHL as described in 2 (Timing if AID in the lymphoma course). The mechanism of AID which occurs before NHL is different from the mechanism of AID which occurs during or after the treatment of NHL. Thus it is important to describe the mechanism of AID in each timing.

In Discussion, authors need to describe about the mechanism of AID in lymphoma. For example, AID which occurs before or during NHL may be associated with the lymphoma cells. However, AID which occurs after chemotherapy might be associated with imbalances of helper/suppressor T cell populations (Oka S, Acta haematologica 2019;141:79-83). The imbalances in T cell functions may lead to the over-reactivity of B cells and autoimmune phenomena.
